# Reasons for and Behavioral Consequences of Male Dog Castration—A Questionnaire Study in Poland

**DOI:** 10.3390/ani12151883

**Published:** 2022-07-23

**Authors:** Marcelina Kriese, Ewelina Kuźniewska, Andrzej Gugołek, Janusz Strychalski

**Affiliations:** Department of Fur-Bearing Animal Breeding and Game Management, University of Warmia and Mazury in Olsztyn, Oczapowskiego 5, 10-719 Olsztyn, Poland; marcelinakriese@gmail.com (M.K.); kuzniewskaewelina18@gmail.com (E.K.); gugolek@uwm.edu.pl (A.G.)

**Keywords:** male dogs, castration, desexing, dog behavior, reasons for castration

## Abstract

**Simple Summary:**

In many Western countries, castration is the most popular surgical desexing procedure in dogs. Castration may deliver health and behavioral benefits, and it is recommended by veterinary and shelter communities. The aim of the present study was to identify the reasons for male dog castration and to determine the owners’ perceptions about changes in dog behavior before and after castration. An online survey was posted on social networking sites dedicated to dogs. The answers showed that the main reason for castration was undesirable behavior, including hyperactivity, roaming, mounting, aggression, marking and others. Castration reduced aggressive behaviors towards dogs and other animals. This surgery increased the number of dogs that were fearful of unfamiliar dogs/humans, as well as dogs with sound phobias, while decreased the prevalence of hiding behavior. Castration greatly decreased incidences of roaming, mounting and urine marking as well as the dog’s overall activity. Thus, it can be concluded that while castration can resolve many undesirable behaviors in male dogs, the arguments for and against neutering should always be considered on an individual basis.

**Abstract:**

In many Western countries, castration is the most popular surgical desexing procedure in dogs. The aim of the study was to identify the reasons for male dog castration and to determine the owners’ perceptions about changes in dog behavior before and after castration. An online survey was posted on social networking sites dedicated to dogs. A total of 386 respondents participated in the survey. The main reason (39%) for castration was undesirable behavior, including hyperactivity (8%), roaming (8%), mounting (7%), aggression (5%), marking (5%) and others (5%). This surgery did not change the prevalence of aggressive behaviors towards people, but it reduced aggressive behaviors towards dogs and other animals. Castration did not reduce the presentation of anxious behavior in fearful dogs. Castration increased the number of dogs that were fearful of unfamiliar dogs/humans, as well as dogs with sound phobias, while decreased the prevalence of hiding behavior. This procedure greatly decreased incidences of roaming, mounting and urine marking as well as the dog’s overall activity. Thus, it can be concluded that while castration can resolve many undesirable behaviors in male dogs, the arguments for and against neutering should always be considered on an individual basis.

## 1. Introduction

Desexing (removal of reproductive organs to prevent reproduction) can be performed using both surgical and non-surgical procedures. In simple terms, surgical desexing is a procedure that involves: (1) castration, namely the surgical removal of the gonads (testicles and epididymides in males; ovaries, oviducts and, in some cases, uteri in females) or (2) sterilization, namely, the ligation of the vas deferens in males and tubal ligation in females. Non-surgical procedures can involve: (1) hormonal, (2) immunological and (3) sclerotization (chemical or physical) methods [1].

Castration is a desexing procedure that has been widely applied in animals since ancient times. In many Western countries, castration is presently the most popular surgical desexing procedure in dogs [1,2]. Castration delivers health and behavioral benefits, and it is recommended by veterinary and shelter communities to control dog populations [3]. However, the empirical evidence in support of castration is not entirely convincing. Research has demonstrated that desexing can reduce the size of the stray dog population, but not the population of companion and shelter dogs [4,5]. With regards to the health consequences of castration in dogs, both indications and contraindications for the procedure have been described [6]. Castration has been found to increase life expectancy by 13.8% in males [7]. Castration removes the testicles and stops the production of sex hormones, which contributes to the prevention of androgen-induced diseases [8]. Castration also reduces the risk of infectious and vascular diseases in male dogs [9]. However, castration in males increases the risk of certain musculoskeletal degenerative diseases and the risk of obesity-related diseases [8,10,11]. Recent studies have shown that there are breed differences in vulnerability to neutering, both with regard to joint disorders (including cranial cruciate ligament rupture and elbow dysplasia) and neoplasia (including osteosarcoma and hemangiosarcoma). Small-dog breeds seemed to have no increased risk of joint disorders associated with neutering, and in only two small breeds (Boston Terrier and Shih Tzu) was there a significant increase in cancers [12]. With regards to the behavioral effects of castration, it seems that desexing generally has a greater impact on the behavior of males than females [13], which could explain why most studies on this topic concern males. It is believed that castration reduces aggression, urinary marking and roaming in males [13,14,15]. However, some studies have shown that castration can increase anxiety levels in dogs and, consequently, the risk of aggressive behavior towards humans [16,17].

Practical considerations aside, canine desexing is also questionable from the ethical point of view. Intact males experience sexual frustration when females are in heat, which also reduces the quality of life of their human owners [18,19]. However, it should also be noted that castration can deprive the male of pleasurable mating experiences [9]. In many countries, desexing is strongly encouraged by veterinarians, while in other countries, routine desexing is considered unethical. Approximately 64% of male and female dogs are neutered in the US, 54% in the UK and 47% in Ireland. In contrast, in Germany and some Scandinavian countries, desexing can be performed only if there are clear medical indications for the procedure [3].

A literature review shows that the behavior of intact and castrated dog populations has been compared in several research studies [3,20,21,22]. However, to the best of our knowledge, the only study analyzing the impact of castration on problematic behaviors in dogs was conducted by Neilson et al. [23] in a small group of 57 dog owners. The aim of the present study was to identify the reasons for male dog castration and to determine the owners’ perceptions about changes in dog behavior before and after castration in a larger group of respondents.

## 2. Materials and Methods

### 2.1. Data Collection

An online survey investigating the behavioral consequences of castration in male dogs was designed and posted on Polish social networking sites dedicated to dogs. The survey contained twenty-three questions (Appendix A). Each survey concerned a single animal. The survey was addressed to the owners whose dogs met the following conditions:The dog was male;The dog was at least six months old on the day of castration;The dog stayed with the current owner for at least six months before castration;The dog had been castrated at least six months before the date of the survey;The dog had undergone surgical castration.

The first part of the survey contained general questions about the respondent’s gender, dog breed/type, dog’s age, age at castration and reasons for castration. The results of the survey were processed by dividing the age at castration and the dog’s current age into the following categories: 6–12 months, 1–2 years, 2–5 years, 5–9 years, >9 years.

The second part of the survey consisted of questions about the impact of castration on dog behavior, including aggression, anxiety, roaming, mounting, over-marking objects and the dog’s activity.

The questionnaires were completed anonymously. All participants gave their informed consent to participate in the survey, and they were informed that the results would be used for research purposes. The survey was conducted between January and April 2021.

### 2.2. Statistical Analysis

The obtained data were analyzed statistically. The χ^2^-test and Fisher′s exact test were used in comparative analyses. Additionally, linear regression analysis was used to analyze the relationship between the age of the dogs (on the day of castration and on the day of the survey) and their behavior after castration. Calculations were performed using the R program [24]. The results were regarded as significant at *p* < 0.05.

## 3. Results

A total of 386 Polish dog owners participated in the survey. The majority of the respondents (91.71%) were female, and males accounted for 8.29% of the respondents.

In the group of 386 analyzed dogs, 145 were crossbreed dogs, followed by Labrador Retrievers (32), German Shepherds (24), Yorkshire Terriers (13), Golden Retrievers (11) and other breeds (Table 1).

Most dogs (118) were castrated at 6–12 months of age (Table 2); 116 dogs were castrated at 1–2 years of age; 92 dogs—at 2–5 years of age; 48 dogs—at 5–9 years of age; and 12 dogs—at >9 years of age. On the day of the survey, most dogs were aged 1–2 years (146 dogs), 2–5 years (130 dogs), 5–9 years (79) and >9 years (31).

The reasons for castration are presented in Figure 1. The main reason (39%) for castration was undesirable behavior, including hyperactivity (8%), roaming (8%), mounting (7%), aggression (5%), marking (5%) and others (5%). In the studied group, 30% of the dogs were castrated due to the owner’s personal conviction that neutering delivers benefits; 17% of the dogs were castrated due to birth control; and 14% of the dogs were castrated due to veterinary recommendations, for example, to prevent disease (7%) or resolve health issues (7%).

Male castration did not induce significant differences in aggressive behavior towards humans (Table 3). In the studied population of 386 dogs, the percentage of dogs that had been aggressive towards humans was 7.51% before castration and 5.70% after castration, and the percentage of dogs that had been sporadically aggressive was 14.51% and 14.25%, respectively. However, neutering decreased the percentage of dogs that were aggressive towards other dogs and other animal species (20.98% vs. 13.99% and 16.06% vs. 10.62%, respectively, *p* < 0.05 in both cases); most respondents reported that after castration, their dogs only displayed aggressive behaviors sporadically.

Castration changed the prevalence of anxious behaviors in dogs (Table 4). Before castration, 13.47% of dogs had been fearful of unfamiliar dogs/humans, and a minor increase in this behavior was observed after castration (18.65%, *p* < 0.05). The number of dogs with sound phobias also increased from 10.62% before castration to 17.10% after the procedure (*p* < 0.01). In contrast, the prevalence of hiding behavior decreased from 19.17% to 11.40% (*p* < 0.01).

Castration significantly decreased the prevalence of roaming, mounting and over-marking behaviors (in each case *p* <0.001; Table 5). The percentage of roaming dogs decreased two-fold after neutering (26.68% vs. 10.61%). The prevalence of mounting behavior (55.44% before neutering) was also halved. The percentage of dogs over-marking objects decreased from 52.59% to 38.86%.

Considerable changes in dog activity were reported after castration (Table 6). Neutering induced a fourteen-fold increase in the percentage of lethargic dogs (from 0.26% to 3.63%, *p* < 0.001), a two and a half-fold increase in the percentage of somewhat active dogs (from 4.66% to 12.69%, *p* < 0.001) and an increase in the percentage of moderately active dogs (from 23.83% to 32.38%, *p* < 0.01). In contrast, castration reduced the percentage of active dogs (from 52.07% to 44.56%, *p* < 0.05) and hyperactive dogs (nearly three-fold, from 19.17% to 6.74%, *p* < 0.001).

Additionally, linear regression analysis revealed the influence of the age of the dogs on the day of castration and on the day of the survey on the prevalence of hiding behavior (*p* = 0.038 and *p* = 0.024, respectively), the age of the dogs on the day of the survey on mounting (*p* = 0.045) and the age of the dogs on the day of the survey on activity level (*p* = 0.018).

## 4. Discussion

The behavior of intact and castrated dogs has been compared in numerous studies. However, this approach has one major drawback, namely, that some dogs may have been castrated due to undesirable behavior. Hence, dogs that were randomly left intact or castrated cannot be identified. The retrospective approach used in the current study also has some limitations, including the possibility that dog owners did not give honest answers when castration did not bring the expected results. Therefore, the present results should be treated with a certain degree of caution.

The percentage of different dog breeds in this study (Table 1) roughly corresponds to the share of dog breeds in the Polish population and is similar to that noted in our recent study [25]. More than 60% of the dogs had been castrated before the age of 2 years (Table 2). The age at castration could be important for health reasons. According to Pollari et al. [26], dogs castrated before the age of 2 years were less likely to experience postoperative problems than older dogs. Puppies and young adults recover more rapidly after castration [27], but every surgical procedure increases the risk of parvoviral infection and hip dysplasia [28]. According to Neilson et al. [23], the age at castration does not correlate with the percentage of improvement in problematic behavior.

In the present study, most dogs were castrated to reduce the prevalence of undesirable behaviors (Figure 1), which is consistent with the findings of other authors [23,29,30]. Maarschalkerweerd et al. [29] observed that the most common behavioral reasons for castration were mounting directed towards people, other dogs and objects, followed by roaming, aggression, marking and fear. In turn, Neilson et al. [23] found that dog owners opted for castration to resolve the following behavioral problems: urine marking, mounting, roaming, aggression toward human family member and aggression toward other dogs in the household. Roulax et al. [30] reported that 58% of the owners neutered their dogs to correct an unwanted behavior, but the types of problematic behaviors were not specified. In our study, dog owners could also give a relatively general reason for castration, namely, personal conviction, and this answer was given by 30% of the respondents. In Poland, castration is widely recommended by veterinarians, printed publications and online articles and during visits to veterinary clinics. Only 14% of the respondents neutered their dogs based on a veterinarian’s recommendation, and half of those did it for health reasons. Only two answers were available in this category: health problems and disease prevention, whereas veterinarians probably also recommended castration for other reasons, such as problematic behavior. A Dutch study demonstrated that dog owners received neutering advice from veterinarians, trainers and therapists, but only veterinary recommendations weighed significantly on their decisions to perform the procedure [30]. Similar observations were made in the UK, where more than four-fifths of the studied population castrated their dogs based on the recommendations made by the veterinarian [9].

As reviewed by D′Onise et al. [31], desexing dogs is, in general, associated with a reduced risk of dog bite. In the current study, castration did not change the prevalence of aggressive behaviors towards people in the studied population, but it reduced aggressive behaviors towards dogs and other animals (Table 3). Similar observations were made by Roulaux et al. [30] who found that neutering decreased the prevalence of general aggressive behavior. However, dog aggression is a complex issue, and aggressive behaviors towards humans and other animals in the household should be analyzed separately from aggression towards strangers. Previous research has shown that castrated dogs are less aggressive towards family members than intact dogs [23], but are more likely to behave aggressively towards unknown people [21]. The latter authors suggested that the age at which the dog is castrated (7–12 months) can influence aggression towards strangers. McGreevy et al. [32] observed that neutering immature dogs decreases the production of gonadal hormones and could increase aggression; consequently, dog owners may not experience this problem if castration is performed in adulthood. The results of the present study suggest that castration at a young age could be associated with an increase in the prevalence of anxious behavior, in particular, the fear of strange dogs/people and sound phobias (Table 4), which corroborates the findings of Farhoody et al. [21]. It should also be noted that fear often causes aggression in dogs. The results presented in Table 4 indicate that neutering does not reduce the presentation of anxious behavior in fearful dogs. However, the prevalence of hiding behavior decreased after castration, which is inconsistent with the above statement. Regression analysis revealed that the decrease in hiding behavior could be attributed, at least in part, to age-related behavioral changes. It is likely that over time, dogs start to feel safe in places they know. As previously mentioned, this observation could also be associated with the owners’ expectations. To the best of our knowledge, this is the first detailed study of hiding behavior in dogs conducted to date; therefore, further research is needed to validate these findings.

Several studies have shown that castration significantly reduces mounting, excessive urine marking and roaming in male dogs [23,29,33], which was fully confirmed by our study (Table 5). However, McGreevy et al. [32] found that urine marking was the only undesirable behavior that was reduced as a result of neutering. They also observed that urine marking is less effectively resolved when a male dog is castrated at an older age due to lifetime exposure to gonadal hormones. Furthermore, they reported an increase in mounting behavior after castration, which was not demonstrated by other studies. Despite the above, a review of the literature and our findings clearly indicate that castration reduces libido and the associated behaviors, such as mounting, excessive urine marking and roaming [3].

A dog′s overall activity profile consists of various behaviors; therefore, previous studies focused on the extent to which castration reduced the prevalence of specific undesirable activities. Neutering appears to diminish a dog’s overall activity levels [3,22], although not all studies support this assumption [34]. Some authors [14,35] have argued that castration decreases activity levels by reducing the frequency of roaming, but according to Heidenberger and Unsheim [13], neutered dogs are less active because they gain weight. In the current study, castration dramatically reduced roaming as well as the dog’s overall activity (Table 5 and Table 6), but body weight was not considered. It is worth mentioning that an important factor influencing the overall activity of dogs is, as the present study showed, the age of the dogs on the day of the survey. Some dog behaviors are characteristic of canines, such as the demand for attention or running, whereas other behaviors are highly undesirable, including pulling a leash, destroying valuable objects and stereotypies. It is worth adding that many undesirable behaviors, especially those that are not directly related to the libido, can be eliminated without castration, for example, by encouraging the dog to engage in other activities that are useful for humans.

## 5. Conclusions

The results of this study indicate that behavioral problems are among the most common reasons for castrating male dogs. Castration reduced the incidence of aggression towards dogs and other animals. Neutering clearly decreased the percentage of dogs engaging in roaming, mounting and excessive urine marking behaviors, and it diminished the dogs’ overall activity levels. Castration did not reduce anxious behaviors in fearful dogs and even increased the number of dogs with a fear of strange dogs/humans and sound phobias. Thus, it can be concluded that while castration can resolve many undesirable behaviors in male dogs, the arguments for and against neutering should always be considered on an individual basis.

## Figures and Tables

**Figure 1 animals-12-01883-f001:**
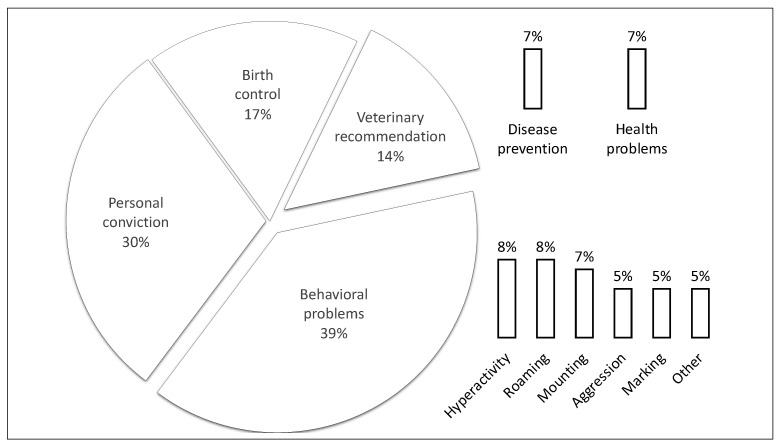
Reasons for dog castration given by the respondents.

**Table 1 animals-12-01883-t001:** Breeds/types of castrated dogs analyzed in the survey.

Breed/Type	Cases ^a^	(*n*) ^b^	Breed/Type	Cases ^a^	(*n*) ^b^
Crossbreed	37.56	(145)	Jack Russel Terrier	2.07	(8)
Labrador Retriever	8.29	(32)	French Bulldog	2.07	(8)
German Shepherd	6.22	(24)	White Swiss Shepherd	2.07	(8)
Yorkshire Terrier	3.37	(13)	Bull Terrier	2.07	(8)
Golden Retriever	2.85	(11)	Beagle	2.07	(8)
Border Collie	2.33	(9)	Other breeds	29.02	(112)

^a^ percentage (%) of answers; ^b^ number of answers.

**Table 2 animals-12-01883-t002:** Dog’s age at castration and dog’s age on the day of the survey.

Age	Age at Castration		Age on the Day of the Survey	
	Cases ^a^	(*n*) ^b^	Cases ^a^	(*n*) ^b^
6–12 months	30.57	(118)	0.00	(0)
1–2 years	30.05	(116)	37.83	(146)
2–5 years	23.83	(92)	33.67	(130)
5–9 years	12.44	(48)	20.47	(79)
>9 years	3.10	(12)	8.03	(31)

^a^ percentage (%) of answers; ^b^ number of answers.

**Table 3 animals-12-01883-t003:** Aggressive behavior towards people, dogs and other animal species before and after castration.

Aggression Towards	Response	Before Castration ^a^	(*n*) ^b^	After Castration ^a^	(*n*) ^b^	*p* ^c^
People	Yes	7.51	(29)	5.70	(22)	0.310
	No	77.98	(301)	80.05	(309)	0.480
	Sporadically	14.51	(56)	14.25	(55)	0.918
Dogs	Yes	20.98	(81)	13.99	(54)	0.011
	No	54.66	(211)	54.15	(209)	0.885
	Sporadically	24.35	(94)	31.87	(123)	0.020
Other animals	Yes	16.06	(62)	10.62	(41)	0.026
	No	67.62	(261)	68.91	(266)	0.699
	Sporadically	16.32	(63)	20.47	(79)	0.137

^a^ percentage (%) of answers; ^b^ number of answers; ^c^ *p*-values in the χ^2^-test against the remaining answers.

**Table 4 animals-12-01883-t004:** Anxious behaviors before and after castration.

Anxious Behaviors	Response	Before Castration ^a^	(*n*) ^b^	After Castration ^a^	(*n*) ^b^	*p* ^c^
Fear of dogs/people	Yes	13.47	(52)	18.65	(72)	0.049
	No	86.53	(334)	81.35	(314)	
Sound phobia	Yes	10.62	(41)	17.10	(66)	0.009
	No	89.38	(345)	82.90	(320)	
Anxiety with aggression	Yes	26.94	(104)	22.02	(85)	0.112
	No	73.06	(282)	77.98	(301)	
Fear of specific objects	Yes	12.44	(48)	11.40	(44)	0.657
	No	87.56	(338)	88.60	(342)	
Freezing behavior	Yes	10.62	(41)	10.62	(41)	1.000
	No	89.38	(345)	89.38	(345)	
Hiding behavior	Yes	19.17	(74)	11.40	(44)	0.003
	No	80.83	(312)	88.60	(342)	
Other	Yes	6.74	(26)	8.81	(34)	0.282
	No	93.26	(360)	91.19	(352)	

^a^ percentage (%) of answers; ^b^ number of answers; ^c^ *p*-values in the χ^2^-test.

**Table 5 animals-12-01883-t005:** Selected undesirable behaviors before and after castration.

Behavior	Response	Before Castration ^a^	(*n*) ^b^	After Castration ^a^	(*n*) ^b^	*p* ^c^
Roaming	Yes	26.68	(103)	10.61	(41)	<0.001
	No	73.32	(283)	89.38	(345)	
Mounting	Yes	55.44	(214)	27.46	(106)	<0.001
	No	44.56	(172)	72.54	(280)	
Urine marking	Yes	52.59	(203)	38.86	(150)	<0.001
	No	47.41	(183)	61.14	(236)	

^a^ percentage (%) of answers; ^b^ number of answers; ^c^ *p*-values in the χ^2^-test.

**Table 6 animals-12-01883-t006:** Activity levels before and after castration.

Activity Rating	Before Castration ^a^	(*n*) ^b^	After Castration ^a^	(*n*) ^b^	*p* ^c^
Lethargic	0.26	(1)	3.63	(14)	<0.001
Somewhat active	4.66	(18)	12.69	(49)	<0.001
Moderately active	23.83	(92)	32.38	(125)	0.010
Active	52.07	(201)	44.56	(172)	0.044
Hyperactive	19.17	(74)	6.74	(26)	<0.001

^a^ percentage (%) of answers; ^b^ number of answers; ^c^ *p*-values in Fisher’s exact test against the remaining answers.

## Data Availability

The data presented in this study are available on request from the corresponding author.

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
