# Peer review of "Reasons for and Behavioral Consequences of Male Dog Castration—A Questionnaire Study in Poland"

_animals, 2022, doi:10.3390/ani12151883_

Round 1

Reviewer 1 Report

Great work to the authors of this paper, it is a greatly needed study to outline the information related to castration. There is minimal I would advise changing, the introduction is well written, the methods are effective and condensed, the results are effective at outlining key information, and finally, they discuss the key information relevant to this area. 

The only area of change I would advise is to possible place table 1 in the appendix, as it takes the reader away from the story and the flow (Which makes the paper less efficient to read). It may be easier to condense into a paragraph? Outlining key information, questions and ordinal and nominal questions.

Overall, great findings! This information is great!

Author Response

Dear reviewer,

thank you very much for reading our manuscript. As you suggested, we moved Table 1 to Supplementary Materials. We also made (after reading other reviews) some minor changes in text which are marked in red.

One more time, thank you!

Reviewer 2 Report

The article provide insights in male dog owners’ perception of their dog’s castration. The subject matter is of interest to readers of Animals. The general design is appropriate for a research article. The article is clearly laid out and all the elements are present. Thus, the prerequisites for a good paper are there. The manuscript, however, lacks detailed data analysis. There are technical errors throughout the manuscript that need correcting.

General comments:

The article was largely fine, although given the paper's focus; I would rephrase the materials and methods with special consideration on the data analysis using linear or binary logistic regression.

Specific comments:

Line 12: retrospective study – the dog owners estimate or know the changes of their dog’s behavior. How did you manage this problem?

Line 14: objectionable change to “undesirable”

Line 18 and 32: urine marking

Line 40: “involves” – check for meaning

Line 84: data collection: It was a survey of Polish dog owners, wasn’t it?

Line 105: With a more detailed statistical analysis, you will get more information.

Line 110: the questions of the survey can present as supplementary materials

Line 112: respondents are … (dog owners of Poland?)

Line 114: Are their different groups of dogs? Are there not analyzed answers?

Line 119 and 135: Table 2 and Figure 1 lack informative description.

Line 125: Table 3 is not necessary, because the whole information are given in the above passage.

Line 163: it is a retrospective study – did the dog owner expected or really observed that the activity of their dog changed, which should be discussed more in detail.

Line 214-216: unclear sentence

Line 221-223: discuss the findings in view of a retrospective study and learned behavior.

Author Response

Dear reviewer,

thank you very much for reading our manuscript. We treated all of your comments with due attention. We have marked the changes in the text in red. All comments are addressed below.

Reviewer wrote:

The article was largely fine, although given the paper's focus; I would rephrase the materials and methods with special consideration on the data analysis using linear or binary logistic regression.

Our response:

- As you rightly suggested, we perform regression analysis. Thanks to this, we expanded the information in the manuscript (Lines 111, 176, 240, 263)

Reviewer wrote:

Line 12: retrospective study – the dog owners estimate or know the changes of their dog’s behavior. How did you manage this problem? Line 163: it is a retrospective study – did the dog owner expected or really observed that the activity of their dog changed, which should be discussed more in detail. Line 221-223: discuss the findings in view of a retrospective study and learned behavior.

Our response:

- The results of retrospective and prospective studies may vary. Both approaches bring some inaccuracies. We refer here to the above remarks in one answer, as they refer to the same thing. The reviewer rightly pointed out that this issue should be commented on, which we do in the new first paragraph of the Discussion (Lines 182-188). Additionally, we provided this issue in the discussion on hiding behavior (Line 242).

Reviewer wrote:

Line 14: objectionable change to “undesirable”

Our response:

- Done.

Reviewer wrote:

Line 18 and 32: urine marking

Our response:

- We corrected it.

Reviewer wrote:

Line 40: “involves” – check for meaning

Our response:

- We checked the meaning of this word in Oxford Thesaurus of English and they give the following examples of the use of this verb: ,,The research involved the assembly of information on unemployment”, ,,I try to involve everyone in key decisions”, ,,Many drug addicts involve themselves in crime”. We consulted our translator and she said that we should not replace this word with another in this context. Well, we propose to keep it.

Reviewer wrote:

Line 84: data collection: It was a survey of Polish dog owners, wasn’t it?

Our response:

- We added the word ,,Polish” in the first sentence of Data Collection. Additionally, we added this information to the title of manuscript.

Reviewer wrote:

Line 105: With a more detailed statistical analysis, you will get more information.

Our response:

- We agree with you. Therefore, we performed the regression analysis mentioned above.

Reviewer wrote:

Line 110: the questions of the survey can present as supplementary materials

Our response:

- We also thought about it. Some authors put the survey questions in the text, and some in the Supplement. After your suggestion, we put the survey in the Supplement.

Reviewer wrote:

Line 112: respondents are … (dog owners of Poland?)

Our response:

- In the first sentence, we changed the word ,,respondents” to ,,Polish dog owners”.

Reviewer wrote:

Line 114: Are their different groups of dogs? Are there not analyzed answers?

Our response:

- We have included this information just in case, as well as information about the country or gender of the respondents. There are some publications on the temperament and suitability for various purposes of selected dog breeds. The more dogs of these breeds, the more likely the chosen behavior may be. We made separated calculations for Labrador Retriever, German Shepherd, Yorkshire Terrier and Golden Retriver breeds, but we had far too few dogs of these breeds to obtain statistical confirmation of any differences. Therefore, we have not provided these more detailed results for publication.

Reviewer wrote:

Line 119 and 135: Table 2 and Figure 1 lack informative description.

Our response:

- We improved it.

Reviewer wrote:

Table 3 is not necessary, because the whole information are given in the above passage.

Our response:

- As you rightly pointed, the same information is given in two places. Since we think the table is easier to read than the text, we have abbreviated the above passage.

Reviewer wrote:

Line 214-216: unclear sentence

Our response:

- We tried to fix it by adding ,,immature dogs” (Line 231).

Once again, we would like to thank the reviewer for all valuable comments. We hope we responded to each comment with due care.

Reviewer 3 Report

In this article, the owners perspective about castration of male dogs was investigated by means of a questionnaire. This questionnaire focused on the reasons for castration and the changes is behavior in terms of aggressive and anxious behavior before and after castration. Despite a relative big amount of respondents (n = 386) and some interesting finding on the population level, I believe that the focus on the individual dog in this study is a missing part.

MAJOR:

-          In the conclusion of the abstract is mentioned that castration of dogs should always be considered on individual basis. However, I miss information about the individual dogs in this study. How many owners report that the behavior resolved (from yes to no, from yes to sometimes)?

-          I miss information about the link with the age. Is there a correlation with the age at the moment of castration and the fact it the undesirable behaviors resolve yes or no ?

-          INTRODUCTION: since this is a study about the effect of castration in male dogs, I don’t see the need to elaborate on the pro’s and con’s of female castration. But I would elaborate a bit more on the recent findings of increased incidence in neoplasia (as osteosarcoma, haemangiosarcoma) and orthopedic problems (as elbow dysplasia, cranial cruciate ligament rupture).

-          DISCUSSION: I miss a part in this discussion about the fact that this information was collected by means of a questionnaire to owners. Are owners capable to see the difference between aggression and fear? What about the ‘placebo effect’ to the owners?

-          DISCUSSION: Mounting behavior, aggression, hyperactivity are all possible coping behaviors because of a chronic stress. I believe that when the dog is not showing these behaviors anymore the owners are happy, but in fact the dog not and probably he will find another way of coping with the chronic stress. Is it possible that dogs change from a more active to a passive coping strategy after castration? Is there any literature available on that? Are there more bite accidents reported after castration?

MINOR:

-          Title: is it possible to add something about the fact that this is a study based on the opinion of owners or a questionnaire.

-          Line 56: what do you mean with a reduction in vascular diseases? Can you give an example?

-          Line 119 & 125: to me, there is no added value of table 1&2

-          Line 138: in the text you mention a reduction in aggressive behavior toward human, however this is not significant. So statistically, there is no reduction.

-          Line 139: idem. Not significant

Author Response

Dear reviewer,

thank you very much for reading our manuscript. We treated all of your comments with due attention. We have marked the changes in the text in red. All comments are addressed below.

Reviewer wrote:

In the conclusion of the abstract is mentioned that castration of dogs should always be considered on individual basis. However, I miss information about the individual dogs in this study. How many owners report that the behavior resolved (from yes to no, from yes to sometimes)?

Our response:

- Thank you for this question. Our conclusion that castration of dogs should always be considered on individual basis was based on the results pointing to 1) a reduction (after the castration procedure) in the frequency of aggressive behavior towards dogs and other animals as well as mounting, roaming and urine marking, 2) an increase in the frequency of some anxious behaviors: fear of dogs/people and sound phobia, and 3) influence of dog age on hiding behavior and mounting (calcuted in new version of manuscript). With regard how many owners reported the behavior resolved agressive behavior towards people: from Y to N 1, from Y to S 7, towards dogs: from Y to N 36, from Y to S 9, towards other animals: from Y to N 5, from Y to S 16.

Reviewer wrote:

I miss information about the link with the age. Is there a correlation with the age at the moment of castration and the fact it the undesirable behaviors resolve yes or no ?

Our response:

- We improved it. In new version of manuscipt we give the results of regression analysis of the age of dogs and their behavior (Line 176).

Reviewer wrote:

INTRODUCTION: since this is a study about the effect of castration in male dogs, I don’t see the need to elaborate on the pro’s and con’s of female castration. But I would elaborate a bit more on the recent findings of increased incidence in neoplasia (as osteosarcoma, haemangiosarcoma) and orthopedic problems (as elbow dysplasia, cranial cruciate ligament rupture).

Our response:

- We deleted information about females castration from Introduction. Regarding neoplasia and orthopedic problems, we found a publication taking into account different breeds of dogs (including their size) and added this information to the text (Lines 59-64).

Reviewer wrote:

DISCUSSION: I miss a part in this discussion about the fact that this information was collected by means of a questionnaire to owners. Are owners capable to see the difference between aggression and fear? What about the ‘placebo effect’ to the owners?

Our response:

- In new version, we added the paragraph in Discussion in which we write about the possibility that dog owners did not give honest answers when castration did not bring the expected results (Line 182-188). We also pointed out that fear is often the cause of aggression (Line 236). We did not ask the respondents what was the cause of their dogs' aggression, so fear could be or not the cause of this behavior.

Reviewer wrote:

DISCUSSION: Mounting behavior, aggression, hyperactivity are all possible coping behaviors because of a chronic stress. I believe that when the dog is not showing these behaviors anymore the owners are happy, but in fact the dog not and probably he will find another way of coping with the chronic stress. Is it possible that dogs change from a more active to a passive coping strategy after castration? Is there any literature available on that? Are there more bite accidents reported after castration?

Our response:

- Thank you for noticing this important thing. We agree that almost everything in life is stress, more or less of course. As you pointed, dogs may choose various coping strategies under the chronic stress. So far, we have not come across a publication that would indicate a reduced physical activity of dogs as a coping strategy after castration. Answering to you second question: a review of the publications indicates that there is general evidence that desexing dogs is associated with a reduced risk of dog bite. We added this information do Discussion (Line 220).

Reviewer wrote:

Title: is it possible to add something about the fact that this is a study based on the opinion of owners or a questionnaire.

Our response:

- Yes, we changed the title.

Reviewer wrote:

Line 56: what do you mean with a reduction in vascular diseases? Can you give an example?

Our response:

- We are sorry, but our research was unsuccessful in this case, no one has elaborated on this information anywhere.

Reviewer wrote:

Line 119 & 125: to me, there is no added value of table 1&2

Our response:

- We have included this information just in case, as well as information about the country or gender of the respondents. There are some publications on the temperament and suitability for various purposes of selected dog breeds. The more dogs of these breeds, the more likely the chosen behavior may be. We made separated calculations for Labrador Retriever, German Shepherd, Yorkshire Terrier and Golden Retriver breeds, but we had far too few dogs of these breeds to obtain statistical confirmation of any differences. Therefore, we have not provided these more detailed results for publication.

Reviewer wrote:

Line 138: in the text you mention a reduction in aggressive behavior toward human, however this is not significant. So statistically, there is no reduction. Line 139: idem. Not significant.

Our response:

- We corrected it (Lines 140-142).

Once again, we would like to thank the reviewer for all valuable comments. We hope we responded to each comment with due care.

Reviewer 4 Report

In my opinion, the work entitled "Reasons for and Behavioral
Consequences of Male Dog Castration" is very interesting. It answers
important questions about the consequences of castration in male dogs, a
procedure that is performed very often in many Western countries. A very interesting aspect of the introduction is to emphasize that castration improves animal welfare.  The article can be published, but below are my minor reservations that should be taken into account:

Line 9 and 45: The word DESEXING (verse 9) does not appeal to me, gender is neurohormonal conditioned, so castration does not result in gender deprivation. "Castration is a desexing procedure" line 45, it sounds bad, in my opinion it would be more accurate to deprive the ability / possibility of reproduction. But maybe it is due to the translation aspects...
Lines 76-77: The authors refer to the review publication [3], but they
should also refer to the original publications on which it is based.
Lines 76-82: The study used a survey method to assess whether
castration, in the opinion of the owners, changed the behavior of male
dogs. They were to evaluate the dog's past and present behavior. As for these abnormal behaviors, a lot also depends on the ability to interpret the dog's behavior. Not every owner can distinguish between typical behavior and misinterprets. Therefore, I would replace the word objectionable behavior with undesirable behavior, because it is often natural behavior of dogs, and only we humans perceive it as inappropriate.
Previous publications relied on the simultaneous comparison of the
behavior of (at the same time) dogs intact vs. castrated. In addition,
the owners also provided reasons why they decided to castrate their
dogs. What I would change in this connection, for the sake of work I
would remove the word "therefore".
When it comes to undesirable behavior, castration does not heal, it only gives us the opportunity to work through certain behaviors with the dog more easily. It is not a "wonder drug". I would say that the surgical castration procedure opens the door to behavioral work on the dog.

I believe that the discussion is well written. The same applies to the
Conclusion.

Author Response

Dear reviewer,

thank you very much for reading our manuscript. We treated all of your comments with due attention. We have marked the changes in the text in red. All comments are addressed below.

Reviewer wrote:

Line 9 and 45: The word DESEXING (verse 9) does not appeal to me, gender is neurohormonal conditioned, so castration does not result in gender deprivation. "Castration is a desexing procedure" line 45, it sounds bad, in my opinion it would be more accurate to deprive the ability / possibility of reproduction. But maybe it is due to the translation aspects... 

Our response:

- We agree with you, however, we would like to explain why we use of this term in relation to dogs’ deprivation of the possibility of reproduction. The reason is that most of the authors we cite use the term ,,desexing”. For example, Urfer et al. 2019 (doi:10.3390/ani9121086) wrote: ,,Desexing is a general term for interventions suppressing fertility in dogs…”, Roulaux et al. 2020 (doi.org/10.1371/journal.pone.0234917) wrote: ,,Desexing dogs regards the surgical removal of the testes in males, more commonly known as castration, or the ovaria in females”. Furthermore, we presented this issue to our translator and she insists that we should leave that word. Therefore, please accept this word in the text. After your suggestion, in the first sentence of the Introduction we try to explain that we mean: ,,Desexing (removal of reproductive organs to prevent reproduction)…”

Reviewer wrote:

Lines 76-77: The authors refer to the review publication [3], but they 
should also refer to the original publications on which it is based.

Our response:

- We added the original publications [20-22].

Reviewer wrote:

Lines 76-82: The study used a survey method to assess whether 
castration, in the opinion of the owners, changed the behavior of male 
dogs. They were to evaluate the dog's past and present behavior. As for these abnormal behaviors, a lot also depends on the ability to interpret the dog's behavior. Not every owner can distinguish between typical behavior and misinterprets. Therefore, I would replace the word objectionable behavior with undesirable behavior, because it is often natural behavior of dogs, and only we humans perceive it as inappropriate.

Our response:

- Thanks to your comment, we changed the word ,,objectionable” to ,,undesirable” throughout the manuscript.

Reviewer wrote:

Previous publications relied on the simultaneous comparison of the 
behavior of (at the same time) dogs intact vs. castrated. In addition, 
the owners also provided reasons why they decided to castrate their 
dogs. What I would change in this connection, for the sake of work I 
would remove the word "therefore".

Our response:

- Thank you for this valuable comment. As you suggested, we deleted the word ,, therefore".

Once again, we would like to thank the reviewer very much for reading our manuscript and adding valuable comments. We hope we responded to each comment with due care. We fully agree with your opinion that ,,castration is not a "wonder drug", it only gives us the opportunity to work through certain behaviors with the dog more easily”.